# Detection System Based on Text Adversarial and Multi-Information Fusion for Inappropriate Comments in Mobile Application Reviews

**Zhicheng Yu \*, Yuhao Jia and Zhen Hong**

The School of Information Engineering, Zhejiang University of Technology, Hangzhou 310023, China;
2112103133@zjut.edu.cn (Y.J.); zhong_zjut@163.com (Z.H.)
\* Correspondence: 2112103127@zjut.edu.cn

**Abstract:** With the rapid development of mobile application technology, the content and forms of comments disseminated on the internet are becoming increasingly complex. Various comments serve as users' firsthand reference materials for understanding the application. However, some comments contain a significant amount of inappropriate content unrelated to the app itself, such as gambling, loans, pornography, and game account recharging, seriously impacting the user experience. Therefore, this article aims to assist users in filtering out irrelevant and inappropriate messages, enabling them to quickly obtain useful and relevant information. This study focuses on analyzing actual comments on various Chinese apps on the Apple App Store. However, these irrelevant comments exhibit a certain degree of concealment, sparsity, and complexity, which increases the difficulty of detection. Additionally, due to language differences, the existing English research methods exhibit relatively poor adaptability to Chinese textual data. To overcome these challenges, this paper proposes a research method named "blend net", which combines text adversarial and multi-information fusion detection to enhance the overall performance of the system. The experimental results demonstrate that the method proposed in this paper achieves precision and recall rates both exceeding 98%, representing an improvement of at least 2% compared to existing methods.

**Keywords:** mobile application; comment detection; text adversarial; multi-information fusion

## 1. Introduction

The rapid development of mobile application technologies has revolutionized people's lifestyles by offering a myriad of convenient functionalities and entertainment options [1]. In this thriving online ecosystem, user reviews serve as the primary source of information for users to understand the practical aspects of various applications [2,3]. However, a significant amount of inappropriate content unrelated to the application itself is being disseminated through these online reviews, including descriptions and links related to pornography, gambling, and other activities [4,5]. These posts not only impact users' browsing experience but also require them to spend more time searching for information that is beneficial to them [6]. Therefore, there is an urgent need for efficient, automated content detection methods. Simultaneously, to optimize user experience, this study categorizes comments unrelated to the app itself, including, but not limited to, pornography, illegal loans, gambling, advertising, and spam content, as inappropriate [7]. Given the sheer volume and complexity of comments on app forums, compiling a training dataset that sufficiently covers all types of inappropriate comments, particularly new ones, presents a challenge [8,9]. Therefore, it is imperative to enhance the intuitive learning and recognition capabilities of detection systems.

Currently, the most advanced deep learning models for comment and sentiment detection are designed and trained primarily for English content, with limited research focused on the Chinese context. Meanwhile, most publicly available comment datasets cur-

rently in use seek to achieve relative category balance during construction [10,11]. In real-world user comment scenarios, app reviews are influenced by various factors such as user preferences and app popularity, resulting in significant disparities in the volume of comments across different categories [12]. This, in turn, leads to poor recognition performance by the model for categories with fewer comments. Moreover, there is a notable absence of Chinese corpora containing inappropriate content [13,14]. The challenge of developing language models capable of identifying inappropriate content in Chinese is exacerbated by its complex nature. Chinese, through its semasiological evolution, is designed to convey multiple layers of meaning, often relying on metaphors and poetic ambiguity. This richness is a testament to its influence, widespread use, and longevity. Conversely, these same characteristics also provide a means for online offenders to evade detection more effectively.

One basic tactic for evading textual detection in Chinese involves substituting standard terms with homophones or internet slang. For example, "米 (Rice)" in "两米包教会注册 (Two Rice Church Registration)" is used as internet slang for "money". Similarly, "佳 (Contact)" and "薇 (WeChat)" in "纯个人出借，急用钱佳我薇***备注苹果通过无前期 (Purely personal lending, urgently need money. Contact me via WeChat *** with the note 'Apple, no upfront fees.)" are phonetic stand-ins for "加 (add)" and "微 (micro)", with their pinyin being 'jiā' and 'wēi', respectively. While similar strategies can be achieved in Western languages, the subtlety and intricacy of the Chinese written language significantly amplify the challenge of detection [15]. Additionally, the scope of inappropriate online Chinese commentary exhibits a substantial topical imbalance. For example, comments pertaining to lending, cash-outs, pornography links, and game account recharging dominate, whereas mentions of invitation code and verification code exchanges are comparatively scarce.

Our contributions to addressing these challenges are highlighted as follows:

- We introduce a novel dataset of Chinese app reviews, marking the first comprehensive and realistic training resource for developing tools to detect inappropriate Chinese reviews.
- We propose a data enhancement strategy using adversarial text to address the imbalance problem and improve model generalizability.
- We offer a multi-information fusion technique that enables developers to leverage the strengths of various deep learning models, thereby increasing detection accuracy and system robustness.
- We develop a standalone model based on Chinese bidirectional encoder representations from transformers (BERTs), presenting a unique solution to the problem of detecting inappropriate Chinese comments.

The remainder of this paper is organized as follows: Section 2 reviews relevant research techniques in this field and identifies existing gaps in technology. Section 3 introduces our experimental methodologies, including the collection and construction of the dataset. Section 4 presents the results of our experiments. Finally, Section 5 concludes the paper with insights and directions for future research.

## 2. Related Works

Traditional methods such as naive Bayes, support vector machines, logistic regression, and decision trees [16,17] have been widely used for detecting toxic comments in online spaces. These approaches, however, struggle with capturing the full spectrum of features in complex information. To address the representation of contextual information, Wang and Zhang proposed a model that integrates a bidirectional gated recurrent unit (Bi-GRU) with a convolutional neural network (CNN) optimized by global pooling [18]. The Bi-GRU component extracts temporal features from reviews, while the CNN employs global pooling for efficient dimensionality reduction. This model culminates in the use of a sigmoid function for outputting classification results. However, enhancing comment feature extraction is only one aspect of the challenge. It is crucial to also consider the correlation between usernames and comment content, as well as content similarities and differences. Focusing

on these factors often means sacrificing the ability to capture nuanced information within the text, resulting in a trade-off between precise classification and model generalizability.

Zhao et al. conducted a study comparing the efficacy of pretrained language models [19] for classifying malicious comments within English-language corpora. Despite the adaptability of these models, their application to Chinese-language data encounters significant barriers. In a different vein, Saumya and Singh introduced a technique combining a long short-term memory network with a CNN and an autoencoder to identify spam comments [20]. Although this approach shows promise in recognizing synonyms and similar linguistic features, it struggles to capture the actual meaning of the content. For example, an innocuous poem sent via email might be incorrectly flagged as spam, whereas a straightforward, unambiguous invitation to commit a crime could slip through unchecked [21]. This underscores the necessity of focusing not only on formal attributes but also on the deep comprehension of the content. Such a dual approach is essential for refining spam detection mechanisms, ensuring they are capable of discerning structural patterns as well as understanding the subtleties of the content [22].

The classification of online reviews focuses on English, leaving a substantial gap in services for Chinese-language forums [23].

The distinct nature of the Chinese language, characterized by its unique grammar, vocabulary, script, and pronunciation, necessitates a detection system specifically tailored for online apps and forum commentary. Addressing the need for a system attuned to Chinese, Zhang and Wang proposed a model that merges a character-level embedded CNN with a Bi-GRU [24]. This innovative approach combines character- and word-level vectors to identify the most important local features within text units. By incorporating a temporal classification method via the Bi-GRU, the accuracy of the model is significantly enhanced. Despite its potential for identifying inappropriate online comments, this method does not tackle the issue of data imbalance highlighted previously. Moreover, the development of the model was constrained by its reliance on a rather limited initial database.

## 3. Methods

### 3.1. Overview

The proposed solution consists of three main steps, as illustrated in Figure 1. Firstly, to address the issue of data imbalance, particularly in identifying comment categories with limited data, we employed an unsupervised text clustering approach. Subsequently, to augment the data volume of these minority categories, we proposed a text enhancement strategy based on sensitive word confrontation. Then, to address the complexity of language and enhance the accuracy and robustness of comment detection, we introduced a multi-information fusion strategy. This strategy consists of multiple modules, where the self-connecting module focuses on sensitive words in comments, the self-capture module captures phrase features of varying lengths within the text, and the interconnect module explores the interrelationships among different texts within comments. Finally, by integrating the extracted textual information, the model's output is converted into a probability distribution through softmax, allowing for the prediction of whether the comment is inappropriate or not.

### 3.2. Text Confrontation Method

#### 3.2.1. Unsupervised Text Clustering of Comment Content Features

Owing to the vast topical diversity found in online app reviews and the general imbalance among these topics, our method initially classifies comments before augmenting the training dataset to improve learning efficiency.

Inappropriate comments often strategically use special symbols to quickly capture users' attention. The critical content is typically positioned close to these symbols, or these symbols are placed near the key content. Table 1 lists some of the most frequently used special symbols for such purposes. Our system employs an emoji library to identify these symbols and captures the nearest 20 characters to mitigate the influence of irrelevant con-

tent and streamline the subsequent semantic analysis. This extracted content is then restructured into a new sentence, facilitating more effective clustering. Table 2 lists various special symbols appearing in this paper, indicating their actual meanings.

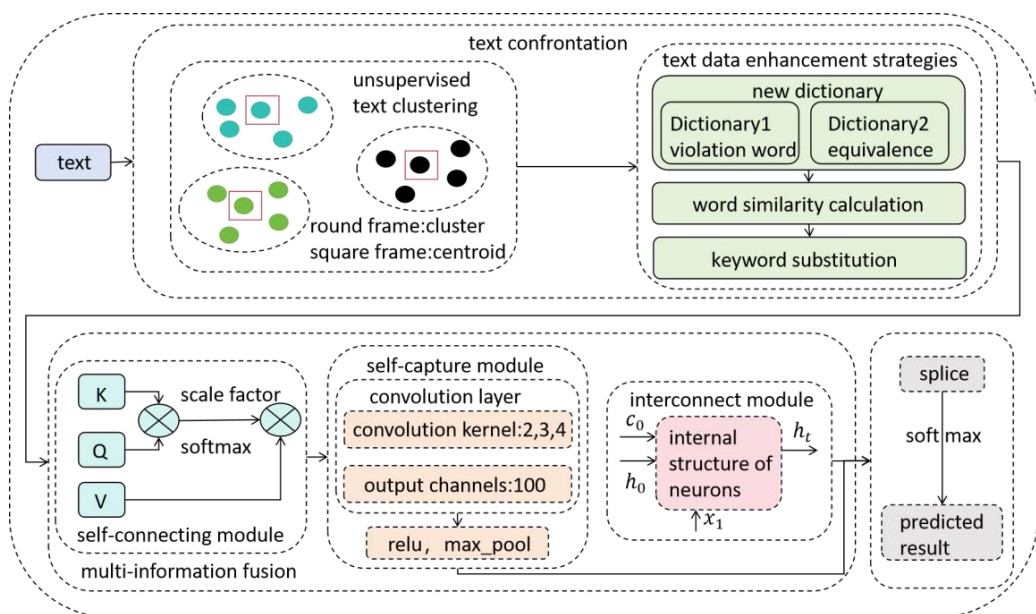

**Figure 1.** Methodological flow of this experiment.

**Table 1.** Special symbols commonly used in inappropriate Chinese online app comments.

| Emoticons | Obvious Punctuation Marks | Other |
| --- | --- | --- |
| 司, 🎋, ☎, 💰 , etc. | +, 《,!!,》 ,● | Two consecutive numeric characters, for example |

In the realm of natural language processing, pretraining the core network has led to significant advancements [25]. A prime example of this is the pretrained BERT model [26], which captures sentence embedding vectors rich in semantic information. This is achieved by first learning from a vast corpus of textual data, allowing the model to grasp the semantics of new statements more effectively. BERT employs a $k$-means clustering method during pretraining, which aids in distinguishing between different categories of content, such as inappropriate content versus compliant content.

This process involves converting sentences into embedding vectors using sample points from the text. $k$-many data points are then randomly selected from the dataset to serve as the initial cluster centroids. The distance between each data point $x_i$ and all centroids is calculated based on their cosine similarity, facilitating the assignment of data points to the nearest cluster. This assignment not only groups the data by proximity but also by semantic similarity, as indicated by the cosine similarity metric:

$$cosine\_similarity\left(x_i, c_j\right) = \frac{x_i \cdot c_j}{\parallel x_i \parallel \cdot \parallel c_j \parallel}. \tag{1}$$

Given $x_i$ as the $i$th sample data point and $c_j$ as the selected centroid, the process for updating the centroids within each cluster involves calculating the mean value of all data points belonging to that cluster to establish the new centroid location. This is achieved by assessing the difference between the previous and newly computed centroids. Through iterative adjustments, data points are reassigned across clusters until there is a negligible change in the centroids' positions, indicating that convergence has been achieved. Additionally, $||x_i||$ and $||c_j||$ represent the lengths of sample data $x_i$ and centroid data $c_j$, re-

spectively. In the calculation of cosine similarity, a similarity value closer to 1 indicates a higher degree of similarity between the two vectors. The centroid updating process is as follows:

$$c_j = \frac{1}{m_j} \sum_{x \in c_j} x, \tag{2}$$

where $c_j$ represents the center of mass for each cluster, updated throughout the iteration process. Here, $m_j$ denotes the number of data points in the $j$th cluster, and $x$ refers to each data point within the cluster.

This methodology underlines the importance of accurately determining the optimal number of clusters ($k$). Choosing $k$ involves balancing the granularity of clustering against the complexity and computational cost of the model. A manual selection of $k$ relies on subjective domain knowledge and experience, which can introduce bias into the training and results. To mitigate this subjectivity and aim for an objective determination of $k$, our system employs the elbow method. This approach involves experimenting with various $k$ values and monitoring the rate of decrease in the sum of squared errors (SSE). As $k$ increases, the SSE typically diminishes, since data points are closer to their respective centroids. However, there is a point at which increasing $k$ further results in a diminishing rate of decrease in SSE, indicating that the optimal number of clusters has been reached. The SSE is calculated using the following formula:

$$SSE = \sum_{i=1}^{k} \sum_{p \in c_i} |x - c_j|^2, \tag{3}$$

where $x$ denotes the data point in each cluster, $c_j$ represents the centroid of each cluster, and $k$ denotes the number of clusters. Through these steps, the system enhances its ability to differentiate among a broader array of text categories.

**Table 2.** The actual meanings of various emoji symbols.

| Emoticons | The Actual Meanings or Interpretations |
|:---:|:---:|
| 可 | can, pronunciation: kě |
| 🐧 | penguin, pronunciation: qǐ é |
| ☎ | telephone, pronunciation: diàn huà |
| 💰 | money, pronunciation: qián |
| ✂ | micro, pronunciation: wēi |
| 🟥 | red envelope, pronunciation: hóng bāo |
| 🌹 | fresh flowers, pronunciation: xiān huā |
| 👉 | point to, pronunciation: zhǐ xiàng |
| 👄 | mouth, pronunciation: zuǐ bā |
| 有 | have, pronunciation: yǒu |
| ❤ | heart, pronunciation: ài xīn |

### 3.2.2. Text Enhancement Strategy Based on Sensitive Word Confrontation

Our sensitive word confrontation technique maintains text semantics by substituting common sensitive words with their equivalents throughout the text. First, Dictionary 1 is created by cataloging common sensitive and inappropriate words. Subsequently, com-

monly used symbols, expressions, and characters are systematically matched with their equivalents to establish Dictionary 2, as illustrated in Table 3.

**Table 3.** Examples of content in Dictionary 1 and Dictionary 2.

| Dictionary 1 | Dictionary 2 |
|---|---|
| 出接 (lend, pronunciation: chū jiē), 薇 (WeChat, pronunciation: wēi), 佳 (Contact, pronunciation: jiā), 福利 (welfare, pronunciation: fú lì), 魏欣 (WeChat, pronunciation: Wèi Xīn), 淇牌 (chess, pronunciation: Qí Pái), etc. | v = 薇 (WeChat, pronunciation: wēi) = 微 (micro, pronunciation: wēi) = 🔀 = 微信 (WeChat, pronunciation: Wēi Xìn), 🔼= 企鹅 (penguin, pronunciation: Qǐ é) = qq, etc. |

For the Chinese characters listed in Dictionary 1, we utilize the lazy_pinyin function from the pypinyin library, an open-source toolkit, to derive the corresponding pinyin for each character. Following this, the Pinyin2Hanzi tool, another open-source resource, is employed to determine the Chinese characters that share the same pinyin. Subsequently, the opencc tool is used to convert these characters into both traditional and simplified Chinese forms, ensuring each character in Dictionary 1 corresponds to a specific set of Chinese characters.

In terms of textual analysis, the jieba tool is applied to eliminate stop words, thereby achieving the final segmentation result. The subsequent step involves verifying if the segmented result is textual; numerical results, for instance, are ignored. Each text participle is then analyzed to determine its stroke count. Segments with a stroke count exceeding four are further divided.

The final stage involves calculating the similarity between the divided words and the entries in Dictionary 1, utilizing the weighted edit distance algorithm. The formula for calculating similarity is as follows:

$$similarity(A, B) = 1 - \frac{D}{max(len(A_1), len(B_1))}, \tag{4}$$

where $A$ and $B$ represent the two words being compared, and $A_1$ and $B_1$ are their respective split results. The weighted edit distance, denoted as $D$, measures the number of modifications required to transition from one character to another. The conversion cost is calculated by dividing the length of the character split by $D$, and this cost is subtracted from 1 to obtain the similarity between the two words. The lower the cost, the higher the similarity.

If the similarity score exceeds 0.65, any word from the group can be used as a replacement for the original word. The replacement of letters, symbols, emoticons, etc., in the comment text is conducted using equivalents from Dictionary 2. The specific criteria for replacement are detailed in Table 4, with examples of replacement outcomes provided in Table 5.

Using these steps, our system generates machine learning-digestible text content with the same context as the original text.

**Table 4.** Specific replacement process.

| Original Word | Original Word Splitting Result | Dictionary Word | Dictionary Word Splitting Result | Similarity | Whether to Replace? | Replacement Word Examples |
|---|---|---|---|---|---|---|
| 薇 (WeChat, pronunciation: wēi) | 艹(cǎo),彳(chì), 山(shān),兀(wù), 攴(pū) | 微 (micro, pronunciation: wēi) | 彳(chì),山(shān), 兀(wù),攴(pū) | 0.8 | yes | 委 (entrust, pronunciation: wěi), 围(surround, pronunciation: wéi) et al. |
| 直播 (live broadcast, pronunciation: zhí bō) | 十(shí),口(kǒu), 二(èr),丨(gǔn), 一(yī),手(shǒu), 丿(piě),米(mǐ), 田(tián) | 直拨 (direct dial, pronunciation: zhí bō) | 十(shí),口(kǒu), 二(èr),丨(gǔn), 一(yī),手(shǒu), 丿(piě),犮(bá) | 0.77 | yes | 纸箔 (foil paper, pronunciation: zhǐ bó), 之播(broadcast, pronunciation: zhǐ bó) et al. |
| 听 (listen, pronunciation: tīng) | 口(kǒu),斤(jīn) | 味 (taste, pronunciation: wèi) | 口(kǒu),一(yī), 木(mù) | 0.33 | no | / |

**Table 5.** Examples of generating new sentences. The symbols "**" and "***" in the table represent random numbers.

| Original Sentence | New Sentence |
|---|---|
| 佳薇Q***23找我领取 力拉 (Add WeChat Q***23, contact me to claim XXX reward) | 家维Q***23找另 力拉 (maintenance Q***23, contact me for another XX reward.) |
| 骚女  acg. **/k2看私密直播 (  Slut  acg. **/k2 Watch Private Live Streaming) | 骚女  acg. **/k2看司迷製播 (  Slut  acg. **/k2 watch fan-made broadcasts) |
| 9***29 QQ 全网比较齐全的返利平台 (9***29 QQ The more complete rebate platform of the whole network) | 9***29 全网比较齐全的范例平台 (9***29 A more complete example of the platform of the whole network) |

*3.3. Multi-Information Fusion*

Upon finalizing the data augmentation process, it is imperative to verify that each category is adequately represented in the dataset. This verification requires the consideration of unique aspects of Chinese characters [27], such as their form and pinyin. The Chinese-BERT pretrained model, enriched with an extensive understanding of Chinese pinyin and glyphs [28], serves as our benchmark for this evaluation.

3.3.1. Self-Connecting Module

The self-connecting module determines the weight values corresponding to text item positions, as illustrated in Figure 2.

First, the text is input and segmented by the pretrained model, with each word subsequently transformed into its corresponding vector. Following this, the importance of each input sequence is determined. From this evaluation, the values *K* (key), *Q* (query), and *V* (value) are derived through a linear transformation of the sequences, represented as

$$K, Q, V = Linear(d_k, d_k), \tag{5}$$

where $d_k$ denotes the dimension of the hidden layer in the pretrained model, specifically 768. These values, $K \in R^{n \times d_k}$, $Q \in R^{m \times d_k}$, and $V \in R^{n \times d_k}$, encapsulate the representation of the input text sequence, the relationship between the query input and label, and the numerical information related to the input sequence, respectively.

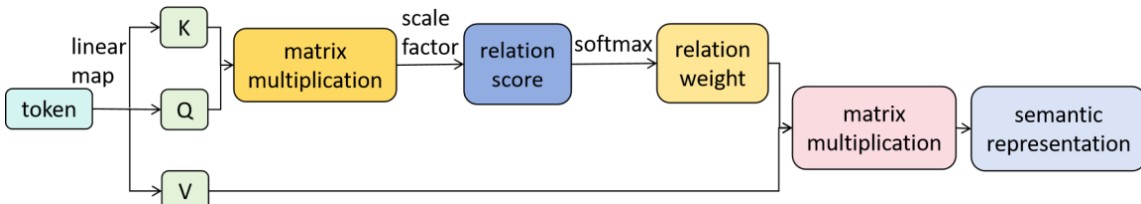

**Figure 2.** Structure of the self-connecting module.

The interaction between $K$ and $Q$ is analyzed to determine which portions of $K$ are focal, determining if the textual content within those segments is crucial for identifying an inappropriate comment. The formula for calculating the correlated segments is as follows:

$$S = \frac{QK^T}{\sqrt{d_k}}, \tag{6}$$

where $S$ represents the relatedness score for a specific portion of text, reflecting its importance in determining the outcome. Given that dot product operations can produce excessively large values, which can potentially destabilize the training process, it is necessary to scale the results by a factor of $\frac{1}{\sqrt{d_k}}$ to normalize the variance.

This scaling ensures that the values at each position are appropriately weighted, allowing the model to selectively concentrate on the most pertinent segments of the input as follows:

$$M = softmax(S), \tag{7}$$

where the *softmax* function serves to transform the scores into a probability distribution, thus ensuring that the association weight M adequately represents the degree of association. This maintains a total sum of 1 during the weighted summation process, guaranteeing numerical stability. Therefore, after normalizing the relationship scores S through the *softmax* operation, a set of association weights M is generated, representing the degree of association between K and Q.

Subsequently, the final semantic representation is obtained. The calculation formula is shown below:

$$A = M \cdot V, \tag{8}$$

where M represents the obtained association weights, which are weighted and aggregated with the corresponding numerical information V to obtain the final semantic representation A.

### 3.3.2. Self-Capture Module

As illustrated in Figure 3, our network architecture is designed to accommodate linguistic features of varying lengths, which is achieved by adjusting the size of the convolutional kernels. This adaptability is crucial for analyzing inappropriate comments, which may include phrases of diverse lengths. Multi-scale convolutional kernels are employed to capture both the local relationships between adjacent words at smaller scales and broader contextual information at larger scales. This dual approach enables our network to attain a more thorough understanding of the text content, facilitating more nuanced and adapt-

able outcomes than might otherwise be attainable. To further refine the analysis, we also employed the relu (rectified linear unit) activation function, which transforms the linear output of the convolutional layer into a nonlinear output. This enables the model to learn and adapt to more complex data distributions, enhancing its generalization capabilities. Simultaneously, MaxPool is introduced, which selects the most prominent features from local regions of the input, effectively reducing the computational load of the model.

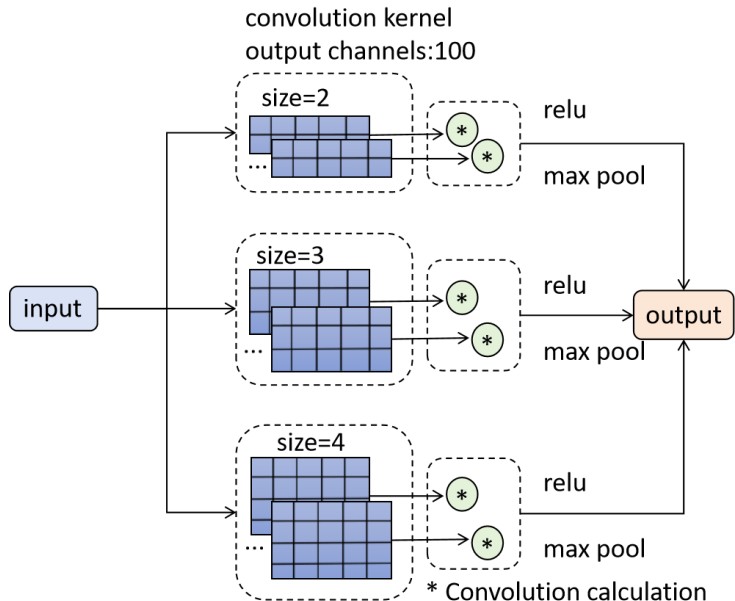

**Figure 3.** Structure of the self-capture module.

### 3.3.3. Interconnect Module

App forums containing inappropriate comments often exhibit a temporal structure, where correlations exist between usernames and the content of the text. This pattern also pertains to the information preceding and succeeding the content. To leverage this interconnectivity, our system introduces a novel interconnectivity module designed to identify and capture long-term temporal information within sequences. This is achieved by recursively updating the internal states of neuron units. The architecture of this module is illustrated in Figure 4, with a detailed view of the neuron unit provided in Figure 5.

First, the comment text is divided into multiple sequences, $X = (x_1, x_2, \ldots, x_t)$, based on its segmentation outcome, with different sequence data being input at various times, facilitating a sequential operation. To mitigate the risks of gradient explosions or vanishing, both the initial state vector $c_0$ and hidden state vector $h_0$ are initialized to zero. The internal structure of the neuron is described by the following formulas:

$$i_t = sigmoid(W_i x_t + U_i h_{t-1} + b_i), \tag{9}$$

$$f_t = sigmoid\left(W_f x_t + U_f h_{t-1} + b_f\right), \tag{10}$$

$$o_t = sigmoid(W_o x_t + U_o h_{t-1} + b_o), \tag{11}$$

where $x_t$ represents the sequence of inputs at different moments $t$, and $i_t$, $f_t$, and $o_t$ correspond to the output values of the input, forget, and output gates within the neuron, respectively. These gates facilitate the control of information flow within the neuron by applying linear transformations to $x_t$ and $x_{t-1}$, followed by the sigmoid function. The weight matrix $W$ connects input $x_t$ to the gate, $U$ is the weight matrix linking $h_{t-1}$ to the gate, and $b$ denotes the bias vector associated with these transformations.

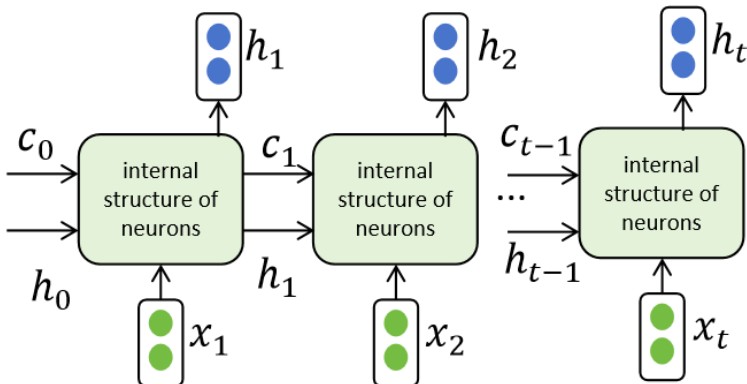

**Figure 4.** Structure of the interconnection module. This module consists of a state vector $c_t$, a hidden state vector $h_t$, an input sequence $x_t$, and a neuronal structure, where t is used to represent different time steps. The state vector $c_t$ is used to store the information that the network has learned over past time steps, facilitating the transmission and updating of this information. The hidden state vector $h_t$ represents the output at the current time step, which is used for making predictions or being passed to the next layer of the network. The input sequence $x_t$ represents the feature vector at the current time step, while the neuronal structure is responsible for processing the sequential data.

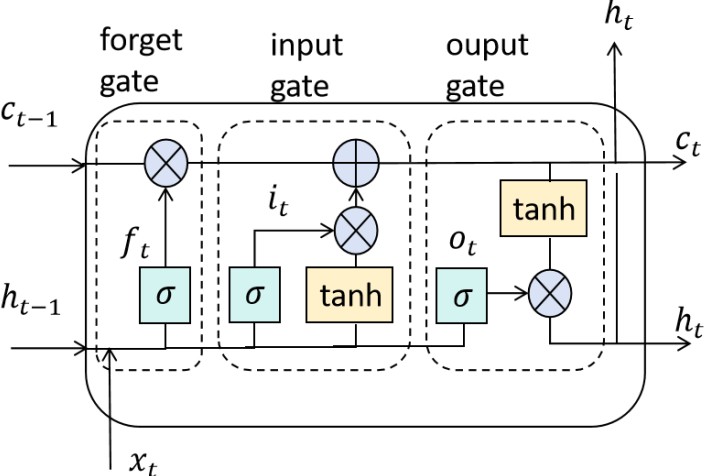

**Figure 5.** Internal structure of the neuron. This neuron consists of a forget gate $f_t$, an input gate $i_t$, and an output gate $o_t$. The forget gate $f_t$ controls which information from the state vector $c_{t-1}$ of the previous time step needs to be forgotten or retained. The input gate $i_t$ determines which information from the current input needs to be injected into the state vector $c_t$. The output gate $o_t$ controls which information from the current state vector $c_t$ should be output, with the outputted information being $h_t$.

The update of the state vector $c_t$ for the current neuron is crucial as it assimilates information from the input sequence. Through continuous updates, this process ensures that valuable information is retained, enabling the capture of semantic connections between sequences. The procedure for updating is as follows:

$$a_t = tanh(W_a x_t + U_a h_{t-1} + b_a), \tag{12}$$

$$c_t = f_t \cdot c_{t-1} + i_t \cdot a_t, \tag{13}$$

where $a_t$ represents the information poised for integration into the state vector $c_t$ of the current neuron. Updating $c_t$ involves selectively forgetting the previous state $c_{t-1}$, selectively incorporating the new information $a_t$ at the current moment, and then combining these two actions. The symbol $c_t$ denotes that information from the previous state $c_{t-1}$ is

incorporated with the current, relevant information $a_t$, thereby updating the neuron with the most up-to-date data.

After computing the state vector $c_t$ for the current moment, the output is generated based on this state vector, resulting in the output vector $h_t$. This vector not only contains information from the current moment but also synthesizes the correlation among the sequences through the integration of $c_t$ information. The calculation of the output vector $h_t$ is expressed as follows:

$$h_t = o_t \cdot tanh(c_t), \tag{14}$$

where, based on the current state information, $c_t$ is selectively output through the output gate of the neuron. Following the extraction of pertinent information, the outputs from both the self-capture and the interconnectivity modules are combined, thereby completing the detection operations.

## 4. Experiment

The primary objective of this study is to assist users in filtering out irrelevant and inappropriate comments on applications, enabling them to efficiently access useful and pertinent information. Prior to conducting the experiment, we need to understand the dataset that will be used. Firstly, Section 4.1 describes the process of data collection and screening for the dataset. Next, in Section 4.2, we introduce the structure of the dataset. Meanwhile, in order to more accurately assess the performance of the proposed method, we elaborate on the dataset partitioning strategies under different experimental conditions in Section 4.3. Subsequently, in Sections 4.4 and 4.5, we present the experimental parameter settings and the evaluation metrics adopted, respectively. In addition, in Section 4.6, we summarize previous related research works and point out their respective strengths and weaknesses. Finally, in Section 4.7, we showcase the final experimental results.

### 4.1. Data Collection

First, a web crawler script was utilized to automate the collection of textual comments directly from the official Apple App Store interface. The focus was on the top 200 apps across four distinct categories: Economy, Games, Shopping, and Social.

During the data collection phase, app review data were continuously gathered from 1 October 2022 to 5 January 2023, spanning a total of 97 days. A cap was set at 200 reviews per day for each application. To capture deleted reviews, an approach was implemented that involved a systematic comparison of comments from consecutive dates. This method allowed for the identification and retrieval of deleted data, considering various scenarios:

1.  For each application, the review data encompass either 200 comments from both the earlier and later periods or fewer than 200 comments in the initial period with the subsequent period reaching 200 comments. Begin by calculating the number of new reviews in the subsequent period for the application and incorporate this into the review data from the earlier period. The review from the earlier period should then be extended in a sequence, and any reviews exceeding the 200 mark, after sorting, are excluded. Subsequently, conduct a one-by-one comparison of this sequentially extended review order against the review data from the earlier period during the subsequent time frame to obtain the deleted reviews.
2.  If the review data for the application are insufficient to reach 200 comments in both the earlier and later time periods, a direct comparison between these periods is feasible. In case the reviews present in the earlier period are missing in the later period, this absence signals their deletion.

The comparison ceases when reviews from the earlier period are not found in the subsequent time frame. These review data are invaluable for examining shifts in user attitudes, platform management strategies, and community relationships.

In summary, we obtained 26,443 user reviews, including 25,142 appropriate entries and 1301 inappropriate entries.

### 4.2. Description of Dataset Structure

The dataset comprises 26,443 review entries, including 25,142 appropriate and 1301 inappropriate entries. Each review in the dataset consists of three core components: username, review subject, and review content. The distribution of review data across various categories is presented in Table 6. This table lists key statistical information such as the number and proportion of total reviews and inappropriate reviews in each category, which helps users quickly understand the overall composition of the dataset.

**Table 6.** Distribution of comment data by data category.

| Category | Total Number of Samples (Items) | Proportion of Total Sample Number | Number of Violation Samples (Items) | Proportion of Illegal Samples |
|---|---|---|---|---|
| economy | 3606 | 13.6% | 169 | 12.9% |
| games | 14,690 | 55.5% | 925 | 71.0% |
| shopping | 4127 | 15.6% | 39 | 3.0% |
| social | 4020 | 15.3% | 168 | 13.1% |

Meanwhile, to facilitate a more intuitive understanding of inappropriate reviews present in each category, we provide examples of inappropriate entries for each category in Table 7.

**Table 7.** Examples of inappropriate entries. The symbols "*", "**" and "***" in the table represent random numbers.

| Economy | Games | Shopping | Social |
|---|---|---|---|
| 只要是苹果都可以借, 不看征信 v: cyt***5 (As long as it is an Apple, you can borrow it without looking at your credit report v; cyt***5) | 现唥\真人\琪牌 ❽❷ *❽. me官网 下裁树木流水 (Planting trees and running water under Xian Xinbi\zhenren\Qipai ❽❷*❽. me official website) | 可+Q13***517套 信 用 卡ieuj (可+Q13***517 set of credit card ieuj) | 骚女 t a** . c n看私密直播翅膀 (Hot girl t a ** . c n watch private live broadcast wings) |
| 纯私人出接 (Purely private pick-up) | 微 175***83 招拖包路费 (Micro 175***83 Recruitment and towing, including tolls) | 《可取花貝jd白條 等》看名字》我很喜 欢先推荐下 (可Take Hua Bei JD Bai Tiao, etc. Look at the name. I like it very much and recommend it first.) | 骚仗 A M AB点©© 看俬密 (Slut Point©©Kanfumi) |

### 4.3. Dataset Division

Text clustering led to the identification of 25 semantic categories. To further distinguish these categories, we have established a criterion: if a category contains less than 33 data entries, it will be defined as "a category with a small number of inappropriate reviews". If a category contains more than 33 data entries, it will be classified as "a category with a significant number of inappropriate reviews". We randomly selected a total of 825 compliant and inappropriate reviews from each category (25 categories × 33 reviews per

category). Among them, 91 belong to the category of "a small number of inappropriate reviews", while 734 belong to the category of "a significant number of inappropriate reviews".

In the experimental design, we considered three different testing scenarios:

1.  Without data augmentation and without introducing the adversarial text component, all data were randomly split into training and testing sets at a ratio of 7:3.
2.  No data augmentation was performed, but the adversarial text component was incorporated. In this case, both the categories with a small number of inappropriate reviews and the categories with a significant number of inappropriate reviews, as well as the compliant reviews, were allocated to the training and testing sets at a ratio of 7:3.
3.  The complete experiment was conducted with data augmentation. For the categories with a small number of inappropriate reviews, they were first split into training and testing sets at a ratio of 7:3. Subsequently, for the reviews assigned to the training set, data augmentation techniques were employed to generate two additional new reviews for each original review, thereby expanding the dataset. The categories with a significant number of inappropriate reviews and the compliant reviews were still split at a ratio of 7:3.

### 4.4. Parameter Setting

The simulation was developed using Python3.7 programming, employing the AdamW optimizer with a learning rate of 0.00001. The weight decay parameter was configured to 0.01, and the cross-entropy loss function was utilized over a total of 50 iterations.

### 4.5. Experimental Performance Indicators

In this paper, we evaluate the performance of the proposed comment detection system for inappropriate comments written in the Chinese language using accuracy and recall as metrics. The formulas for these indicators are as follows:

$$accuracy = \frac{TP + TN}{TP + TN + FP + FN},\tag{15}$$

$$recall = \frac{TP}{TP + FN},\tag{16}$$

where TP represents true positives, TN denotes true negatives, FP stands for false positives, and FN signifies false negatives.

### 4.6. Baseline Experimental Model

In this section, we introduce the baseline experimental model approach used, along with its advantages and disadvantages.

BERT [26]: A bidirectional encoder representation model based on a transformer that captures contextual information in text through multiple layers of self-attention mechanisms and attention weights. This model learns a significant amount of linguistic knowledge during the pre-training phase, demonstrating strong text representation capabilities and versatility. However, BERT's pre-training dataset primarily consists of large English corpora such as Wikipedia, with relatively limited support for other languages. This can affect its performance in non-English tasks.

RoBERT [29]: This model adopts a similar architecture to BERT during the pre-training phase, namely, a bidirectional encoder based on a transformer. RoBERT achieves improvements through the use of larger datasets, longer pre-training durations, and optimization strategies such as adjusting batch size and sequence length. However, similar to BERT, RoBERT also focuses primarily on English text during pre-training and may not adequately consider the unique characteristics of Chinese text. Therefore, when dealing with complex Chinese text, RoBERT may require additional optimization to achieve optimal performance.

Chinese-BERT [28]: This model incorporates linguistic features specific to Chinese by integrating character glyphs and pinyin information during pre-training, aiming to en-

hance its text semantic understanding capabilities. This integration helps the model better understand the semantic information of Chinese text. However, despite its enhanced performance in Chinese tasks, Chinese-BERT may still face some challenges when dealing with specific types or complex structures of Chinese text. For example, when identifying deep-level linguistic phenomena such as metaphors or puns in reviews, relying solely on character glyphs and pinyin information may not be sufficient to fully comprehend the true meaning of the text.

gzip [30]: This method combines a compressor with a k-nearest neighbor classifier to address the complexity of parameters when training neural network models. However, when dealing with reviews containing complex text content, this method may face limitations in generalization ability, making it unable to fully capture deep semantic information within the text.

*4.7. Experimental Outcomes*

4.7.1. Comparison Experiment

To validate the effectiveness of various methods, we conducted comparative experiments across three different scenarios, all originating from identical baseline conditions. In the experiments presented in Table 8, the text adversarial component was omitted. Here, data were randomly sampled before conducting tests on the multi-information fusion component. The experiments presented in Table 9 incorporated text clustering but did not apply data augmentation to the smaller sample sets. Subsequently, tests on the multi-information fusion component were executed. Simultaneously, Table 10 presents the comprehensive experiment suite, where experiments involving multi-information fusion were performed after data augmentation via text adversarial techniques. A comparison of the results in Tables 9 and 10 demonstrates the effectiveness of our text clustering method. In the absence of text clustering, inappropriate comment data were extracted randomly, mirroring this randomness in their distribution. This increased the ambiguity in feature detection, leading to diminished accuracy and recall rates for compliant comments in comparison to those for inappropriate comments. Introducing text clustering helps reduce the uncertainty of inappropriate comment data, improves the efficiency of capturing inappropriate comment features, and subsequently enhances the system's performance in inappropriate comment detection tasks.

A comparison between Tables 9 and 10 demonstrates the effectiveness of our data enhancement technique. Before applying data augmentation, there was a notable imbalance in the data volume within the inappropriate comment category, where "inappropriate" constituted the minority category. Addressing this imbalance is essential for enhancing both accuracy and recall.

The analysis of Tables 8–10 reveals that our approach offers distinct advantages over alternative methods. Specifically, although large-scale pre-trained models like BERT and RoBERT have accumulated a significant amount of linguistic knowledge during the pre-training phase, they primarily rely on English datasets for training. As a result, when dealing with Chinese tasks, they may not adequately consider the unique characteristics of Chinese text, such as character glyphs and pinyin. This, to some extent, limits their performance in Chinese text processing. Meanwhile, Chinese-BERT attempts to address this limitation by incorporating character glyph and pinyin information into the pre-training of the language model, thereby enhancing performance to some extent. However, compared to blend net, Chinese-BERT still lags slightly in terms of accuracy and recall. This suggests that, in addition to character glyphs and pinyin information, there are other critical factors (such as sensitive word information, phrase features, contextual information, etc.) that are crucial for understanding Chinese text. Furthermore, while gzip contributes to addressing the complexity of parameters during neural network model training, it may not fully capture deep-level linguistic features when processing Chinese text. The method proposed in this paper achieves improved performance in detecting inappropriate com-

ments in Chinese text by integrating information related to pinyin, character shapes, and contextual relationships.

**Table 8.** Performance without text confrontation.

| Methods | Accuracy | Recall |
|---|---|---|
| BERT [26] | 0.893 | 0.916 |
| ROBERT [29] | 0.871 | 0.910 |
| Chinese-BERT [28] | 0.923 | 0.936 |
| gzip [30] | 0.906 | 0.913 |
| blend net (our method) | 0.940 | 0.953 |

**Table 9.** Performance without data enhancement.

| Method | Accuracy | Recall |
|---|---|---|
| BERT [26] | 0.930 | 0.919 |
| ROBERT [29] | 0.926 | 0.773 |
| Chinese-BERT [28] | 0.940 | 0.937 |
| gzip [30] | 0.937 | 0.910 |
| blend net (our method) | 0.961 | 0.954 |

**Table 10.** Performance with data enhancement.

| Method | Accuracy | Recall |
|---|---|---|
| BERT [26] | 0.950 | 0.931 |
| ROBERT [29] | 0.933 | 0.767 |
| Chinese-BERT [28] | 0.969 | 0.966 |
| gzip [30] | 0.960 | 0.929 |
| blend net (our method) | 0.984 | 0.988 |

4.7.2. Ablation Experiments

To validate the individual contributions of each module to the overall effectiveness, we conducted three ablation studies under identical conditions. These ablation experiments, detailed in Tables 11–13, replicated the experimental setup of Tables 8–10, respectively. Specifically, Table 11 outlines experiments where the text adversarial component was omitted; here, data were randomly sampled prior to conducting analyses of the multi-information fusion component. In the experiments presented in Table 12, text clustering was utilized without the application of data augmentation, followed by analyses of the multi-information fusion component. Table 13 presents the complete experimental sequence, encapsulating the comprehensive methodology employed.

Through a comprehensive analysis of the experimental results in Tables 11–13. Firstly, we can observe that in all three experimental scenarios, integrating the information provided by each module enhances the performance of the model. This indicates that each module contributes to performance improvement, and their integration allows for a more comprehensive capture of comment features, resulting in more powerful text processing capabilities. Specifically, the self-connecting module focuses on sensitive words within comments, the self-capture module captures phrase features of varying lengths in the text, and the interconnect module explores the interrelationships between different texts within the comments. The synergistic effect of these modules enables the model to process complex Chinese text more accurately and efficiently.

**Table 11.** Our model without text clustering.

| Methods | Accuracy | Recall |
|---|---|---|
| base model + interconnect module | 0.924 | 0.939 |
| base model + self-connecting module | 0.928 | 0.938 |
| base model + self-capture module | 0.926 | 0.938 |
| base model + interconnect module + self-capture module | 0.936 | 0.941 |
| base model + interconnect module + self-connecting module | 0.931 | 0.944 |
| base model + self-capture module + self-connecting module | 0.934 | 0.951 |
| base model + interconnect module + self-capture module + self-connecting module (our method) | 0.940 | 0.953 |

**Table 12.** Our model without data enhancement.

| Methods | Accuracy | Recall |
|---|---|---|
| base model + interconnect module | 0.942 | 0.942 |
| base model + self-connecting module | 0.950 | 0.940 |
| base model + self-capture module | 0.944 | 0.939 |
| base model + interconnect module + self-capture module | 0.959 | 0.945 |
| base model + interconnect module + self-connecting module | 0.952 | 0.949 |
| base model + self-capture module + self-connecting module | 0.957 | 0.951 |
| base model + interconnect module + self-capture module + self-connecting module (our method) | 0.961 | 0.954 |

Additionally, by comparing the results under different experimental conditions, we can observe the impact of using text clustering and data augmentation on improving model performance. This indicates that the issue of data imbalance has been effectively addressed through the data augmentation of a small number of class samples.

**Table 13.** Our model using data enhancement.

| Methods | Accuracy | Recall |
|---|---|---|
| base model + interconnect module | 0.970 | 0.975 |
| base model + self-connecting module | 0.973 | 0.972 |
| base model + self-capture module | 0.971 | 0.968 |
| base model + interconnect module + self-capture module | 0.982 | 0.977 |
| base model + interconnect module + self-connecting module | 0.976 | 0.981 |
| base model + self-capture module + self-connecting module | 0.979 | 0.983 |
| base model + interconnect module + self-capture module + self-connecting module (our method) | 0.984 | 0.988 |

Furthermore, this method achieves optimal performance when integrating information from multiple modules and utilizing data augmentation. This result demonstrates the effectiveness of our model design, which can fully utilize various rich information to improve the accuracy and efficiency of Chinese text processing. This provides strong support and references for future research and applications in related fields.

## 5. Conclusions and Future Research

In this paper, we conducted an in-depth study on increasingly complex user comments regarding applications. These comments are often interspersed with a significant amount of inappropriate content unrelated to the application itself, such as gambling, lending, pornography, and top-ups, which severely affect the user experience. To address this issue, we designed a comment content detection method called "blend net" that is independent of the application. This method aims to help users filter out irrelevant and inappropriate messages, enabling them to quickly obtain useful and relevant information. Firstly, considering the sparsity of data, where the number of comments in certain undesirable categories is relatively small, we propose a sample generation method based on textual adversarial techniques. Through unsupervised text clustering, we can effectively identify minority class data within comments and utilize sensitive word adversarial strategies to augment these limited class sample data, thereby enhancing the generalization ability of the model. Secondly, we designed a multi-information fusion approach to address the complexity of comment texts. This method uses Chinese-BERT as the baseline model and combines three modules: a self-connecting module, a self-capture module, and an interconnect module. By integrating pinyin and glyph information of Chinese text, focusing on sensitive vocabulary, exploring text features, and capturing contextual connections between texts, this method improves the accuracy and robustness of recognition. Finally, through experimental evaluation, the accuracy and recall rate of the method proposed in this paper have reached over 98%. Compared with existing methods, it achieves at least a 2% improvement in performance. This result demonstrates the effectiveness and advancement of the proposed method.

Furthermore, for future work, we plan to make improvements in the following aspects: 1. We will increase the number of inappropriate Chinese comments considered, collect a wider variety of comment forms and styles, and enhance the difficulty of detection. 2. More applicable data enhancement methods are needed to improve model generalizability and robustness in practical applications. 3. Applying additional and more challenging experimental conditions will help to accurately evaluate the accuracy and recall of this system.

**Author Contributions:** Conceptualization, Z.Y.; methodology, Z.Y. and Y.J.; software, Z.Y.; validation, Y.J.; formal analysis, Z.H.; investigation, Z.Y.; resources, Z.H.; data curation, Y.J.; writing—original draft preparation, Z.Y. and Y.J.; writing—review and editing, Z.H.; visualization, Y.J.; supervision, Y.J. and Z.H.; project administration, Z.H.; funding acquisition, Z.H. All authors have read and agreed to the published version of the manuscript.

**Funding:** This research was funded by the National Natural Science Foundation of China, grant number 62072408.

**Institutional Review Board Statement:** While responsibly sharing our findings with the relevant app markets, particularly the Apple App Store, we have not yet received any response from them. Throughout our research, we have consistently prioritized ethical considerations and adhered to established protocols to safeguard the rights and well-being of all parties involved.

**Data Availability Statement:** The datasets we used and analyzed during the current study are available from the corresponding author upon reasonable request. Simultaneously, we only gather publicly available comments from the official interface of the Apple App Store, ensuring no privacy concerns are involved. We guarantee that these data are exclusively utilized for research purposes.

**Conflicts of Interest:** The authors declare no conflicts of interest.

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
