# Peer review of "Detection System Based on Text Adversarial and Multi-Information Fusion for Inappropriate Comments in Mobile Application Reviews"

_electronics, doi:10.3390/electronics13081432_

Round 1
Reviewer 1 Report
Comments and Suggestions for Authors
The authors are proposing latent semantic analysis, a method that already exists. Why is the variation novel? The authors should clarify the difference between mobile and other content.
The dataset should measure and explain why just a subset of emoticons is studied. Finally, additional experiments are needed.
The English should improve in all the manuscripts.
Author Response
I am deeply grateful to the reviewer for posing invaluable questions, which have been immensely helpful to my research. In order to provide a more detailed response to your inquiries, I have organized my replies into a document and attached it for your review. Thank you once again for your thoughtful examination and guidance. I look forward to receiving further feedback from you.

Reviewer 2 Report
Comments and Suggestions for Authors
This study aimed to detect inappropriate comments in Mobile applications based on text adversarial and multi-information fusion. The paper is well presented, written with care, and the overall structure makes good sense. However, some issues should be resolved:
1- A full description of the dataset structure should be given.
2- I recommend you add a summary table for the previous work, methods used in such work, advantages, and weaknesses.
3- The authors mentioned data augmentation, but they didn’t mentioned the size of the data before and after the augmentation process.
4- The Adam optimizer has been used in the study, why? Have the authors tried any other optimizer?
5- Is there any particular reason why the authors choose softmax and cross-entropy?
Author Response

(The authors gave the same response as above.)

Reviewer 3 Report
Comments and Suggestions for Authors
The paper "Detection System Based on Text Adversarial and Multi-infor-2 mation Fusion for Inappropriate Comments in Mobile Applica-3 tions" analyzes the real comments of various Chinese applications in the Apple App Store and highlights the irrelevant comments a certain degree of obscurity, rarity and complexity. Also, due to language differences, existing research methods in English present relatively poor adaptability to Chinese textual data. The present study proposes a research method that combines contradictory text with more detection information to improve the overall performance of the system. Experimental results demonstrate that the method proposed in this paper achieves precision.
To improve your work, you should consider the following aspects:
1. In the first paragraph, the text "The rapid development of mobile application (app) technology..." must be written in the plural, because there are many mobile technologies.
2. Rewriting the sentence "Finally, our analysis yielded 26,443 comment excerpts", replacing the term "Finally"...line(361)
3. The relationship between the elements of figure 1 is not shown.
4. In formula 1 the term cosine is written with spaces between the letters. Not all terms of formula 1 are explained.
5. Column 2 in table 2 exceeds the page.
6. The formula 7 is written incorrectly and must be explained, respectively a citation regarding softmax should be used.
7. In figure 3, it would be good to explain that ReLu means "rectified linear activation function".
8. The terms in figures 4 and 5 must be explained.
9. Attention: between October 1, 2022 and January 5, 2023, there are 97 days NOT 67 (line 340).
10.The results from tables 5-10 are very briefly analyzed, correlated and interpreted.
11. The conclusions must be detailed.
12. Lines 21,22,23..The experimental results demonstrate ..they are identical to the lines 445,446,447.
moderat.
Author Response

(The authors gave the same response as above.)

Round 2
Reviewer 1 Report
Comments and Suggestions for Authors
The researchers conducted experiments to answer their research questions. Additionally, Chinese symbols need to have an English equivalent.
Comments on the Quality of English LanguageSome minor errors should be corrected. Additionally, the authors should add a paragraph between each section.
Author Response
Dear reviewer, I am deeply grateful for your valuable questions, which are very helpful to my research.In order to provide a more detailed response to your inquiries, I have organized my replies into a document and attached it for your review. Thank you once again for your thoughtful examination and guidance. I look forward to receiving further feedback from you.
